

# Evaluation of reanalysis precipitable water vapor under typhoon conditions using multi-source observations

Jiaqi Shi[1,2], Min Li[1,2], Andrea Karin Steiner[3], Sebastian Scher[3,4], Minghao Zhang[5], Jiayu Hu[1,2], Wenliang Gao[1,2], Yongzhao Fan[1,2], and Kefei Zhang[5]

5 [1]GNSS Research Center, Wuhan University, Wuhan, 430079, China
[2]School of Geodesy and Geomatics, Wuhan University, Wuhan, 430079, China
[3]Wegener Center for Climate and Global Change, University of Graz, 8010 Graz, Austria
[4]Department of Geography and Regional Sciences, University of Graz, 8010 Graz, Austria
[5]School of Environmental Science and Spatial Informatics, China University of Mining and Technology, Xuzhou 221116, 
10 China

*Correspondence to*: Min Li (limin@whu.edu.cn)

**Abstract.** Precipitable water vapor (PWV) exhibits rapid and complex variations during typhoons, and its evaluation under typhoon conditions remains challenging due to sparse observations over the oceans. This study systematically evaluates PWV estimates from three state-of-the-art reanalyses during 113 typhoons between 2020 and 2024 over the Northwest 15 Pacific and East Asia. The fifth-generation European Centre for Medium-Range Weather Forecasts Reanalysis (ERA5), the Modern-Era Retrospective Analysis for Research and Applications Version 2 (MERRA-2), and the Japanese Reanalysis for Three Quarters of a Century (JRA-3Q) are compared with ground-based Global Navigation Satellite System (GNSS), radiosondes, and radio occultation (RO) observations. Results based on ground-based GNSS show that ERA5 provides the highest accuracy, with a bias of −1.65 mm in non-typhoon periods and an even smaller bias of −0.29 mm during typhoons, 20 and root mean square error (RMSE) decreasing from 3.52 mm in non-typhoon periods to 3.40 mm during typhoons. JRA-3Q also has smaller error during typhoons compared to non-typhoon periods though its bias and RMSE remain relatively large. Conversely, MERRA-2 shows higher error during typhoons compared to non-typhoon periods, shifting from a modest underestimation of −0.53 mm in non-typhoon periods to an overestimation of 0.86 mm during typhoons, but still maintains accuracy throughout typhoon periods. PWV estimates from all three reanalyses show high correlations with those from 25 radiosonde and RO observations. These results provide a comprehensive accuracy reference and confirm the suitability of reanalyses for PWV researches during typhoons, with ERA5 appearing the most reliable among the datasets evaluated.

## 1 Introduction

Water vapor is primarily distributed within the troposphere and is one of the most important greenhouse gases. It plays a critical role in energy exchange within weather systems, the hydrological cycle, and climate change (e.g., Schneider et al., 30 2010; Sherwood et al., 2010). Precipitable water vapor (PWV), defined as the total amount of water vapor contained in a vertical column of the atmosphere per unit area, is a key variable for characterizing atmospheric water vapor. It is widely



used in meteorological and climatological monitoring and forecasting (e.g., Zhao et al., 2020; Zhang et al., 2022). The spatio-temporal variation and distribution of PWV does not only influence the vertical humidity structure and water vapor transport processes but are also closely associated with the formation and development of various extreme weather events,

including severe convective systems (Kim et al., 2022; Liu et al., 2023).

Tropical cyclones (TCs) are among the most destructive types of extreme weather. They occur frequently, exhibit high intensity, and cause widespread impacts (Emanuel, 2005; Walsh et al., 2012; Chan et al., 2018; Wang et al., 2020; Shi et al., 2021; Xi et al., 2023). TCs are often accompanied by heavy rainfall and secondary disasters such as flooding, landslides, and debris flows (Woodruff et al., 2013; Cogan et al., 2018; Utsumi and Kim, 2022). With intensifying global warming, the

average translational speed of TCs has decreased by approximately 10%, while their associated precipitation has increased by about 15%, leading to more prolonged impacts in affected regions (Elsner, 2020; Intergovernmental Panel on Climate Change (IPCC), 2022; Tran et al., 2022).

TCs occurring over the Northwest Pacific and the South China Sea are referred to as typhoons. In recent years, typhoons have frequently struck countries in the Asia-Pacific region, including China, South Korea, and Japan, with southeastern

China being particularly vulnerable. These events have resulted in substantial casualties and economic losses (Esteban and Longarte-Galnares, 2010; Jung et al., 2024; Wang et al., 2024). Some intense typhoons and their residual circulations have even penetrated deep into inland China. For example, Typhoon In-Fa (2106) in 2021 and the double typhoons Doksuri (2305) and Khanun (2306) in 2023 caused severe flooding in northern and northeastern China (Shi et al., 2022; Zhao et al., 2024). As a type of extreme weather driven in part by atmospheric moisture, typhoons are strongly coupled with the spatio-temporal

distribution of PWV. PWV reveals the pathways and intensity of moisture transport during typhoons and exhibits a strong physical response to their evolution and movement. Therefore, high-accuracy PWV estimates are crucial for understanding the mechanisms underlying extreme precipitation during TCs, improving TC monitoring and forecasting, and supporting disaster risk assessment and mitigation efforts.

Various measurement techniques have been employed to retrieve PWV, including radiosondes, water vapor radiometers,

satellite-based microwave/infrared remote sensing, and sun photometers (Ichoku et al., 2002; King et al., 2003; Li et al., 2003; Turner et al., 2007). In recent decades, the development of Global Navigation Satellite System (GNSS) technologies and Low Earth Orbit (LEO) satellites has enabled the widespread application of ground-based GNSS and space-based GNSS radio occultation (RO) for atmospheric observations (e.g., Melbourne et al., 1994; Kursinski et al., 1996; Li et al., 2017). However, these techniques have limitations, especially when applied to fast-moving, ocean-based, and moisture-complex

weather systems such as TCs. Accurately estimating PWV with both high resolution and temporal continuity remains challenging.

Conveniently, gridded global reanalysis datasets provide atmospheric fields with high spatial and temporal resolution and no gaps, making it possible to obtain PWV at any time and location through interpolation. Reanalysis data thus offer valuable resources for retrospectively investigating moisture transport and evolution during typhoons and for characterizing the

spatial and temporal features of water vapor. Currently, several research centers provide atmospheric reanalysis datasets that



are widely used, including the fifth-generation European Centre for Medium-Range Weather Forecasts (ECMWF) Reanalysis (ERA5) (Hersbach et al., 2020a), the Modern-Era Retrospective Analysis for Research and Applications Version 2 (MERRA-2) (Gelaro et al., 2017a), and the Japanese Reanalysis for Three Quarters of a Century (JRA-3Q) (Kosaka et al., 2024).

Due to differences in data assimilation strategies and the uneven spatio-temporal distribution of assimilated observations, the accuracy of PWV estimates from reanalysis datasets remains uncertain. Therefore, we perform a systematic and comprehensive evaluation to assess the accuracy and applicability of PWV products from reanalyses prior to practical application.

Ground-based GNSS-PWV, with high accuracy of typically within 1–2 mm, is not assimilated into any of the three
reanalysis datasets examined in this study, making them independent and reliable reference data for validation (Wang et al., 2020; Li et al., 2025). However, given the sparse coverage of GNSS stations over oceans, many studies have also employed radiosonde and GNSS radio occultation (RO) observations as complementary validation sources.

At the regional scale, evaluations have shown that ERA5 achieves generally lower PWV errors (<1 mm) over China, outperforming its predecessor ERA-Interim, with cross-validation using radiosonde data further confirming its reliability
(Zhang et al., 2019; Zhang et al., 2019). In India, ERA5 also clearly outperforms MERRA-2 in PWV monitoring (Rani and Singh, 2025). Over the southern Tibetan Plateau, multiple reanalysis products exhibit systematic positive biases in the seasonal PWV cycle, likely linked to the persistent wet bias in regional models (Wang et al., 2017). In the Arctic, evaluations suggest that the Copernicus Arctic Regional Reanalysis (CARRA) provides accurate PWV estimates and shows good agreement with radiosonde observations, although with evident seasonal variability (Zhang et al., 2025). At the global
scale, reanalysis PWV has shown good agreement with GNSS, radiosonde, and RO observations (Zhang et al., 2018). However, considerable uncertainties remain in tropical and southern hemisphere regions, particularly in PWV estimates from the National Center for Environmental Prediction/Department of Energy (NCEP/DOE) dataset (Vey et al., 2010). Moreover, geographic and climatic factors have been shown to influence the consistency between reanalysis and GNSS-derived PWV (Bock and Parracho, 2019). Among the various products, ERA5 generally outperforms others, while China's newly released
global reanalysis dataset, China Meteorological Administration-40 (CRA40), exhibits comparable performance to ERA5 in PWV estimation (Wang et al., 2020; Li et al., 2025), However, as CRA40 data are not fully publicly available, it is not included in this study. GNSS RO data have also been used to compare ERA5 and MERRA-2 PWV estimates in tropical and subtropical regions (Johnston et al., 2021). Additionally, recent studies have identified humidity modeling discontinuities in ERA5 at 09:00 and 21:00 UTC, which introduce diurnal jumps in zenith tropospheric delay (ZTD) and subsequently affect
PWV estimations (Yuan et al., 2025).

Existing evaluation studies have primarily focused on long-term and large-scale averages, with limited systematic assessment of PWV accuracy from reanalysis datasets under extreme weather conditions such as typhoons. Moreover, comparative analyses of PWV estimation accuracy between typhoon and non-typhoon periods remain scarce.





Driven by this research gap, we provide a systematic evaluation of PWV estimates from ERA5, JRA-3Q, and MERRA-2
during typhoon events using ground-based GNSS, radiosonde, and RO data from January 2020 to December 2024 over the
Northwest Pacific and East Asia region. The data and methods used in this study are introduced in Section 2. Section 3
presents the results, and Sections 4 and 5 provide a discussion and conclusions, respectively.

2 Data and methodology

## 2 Data and methodology

In this section we introduce the typhoon datasets, the three reanalyses, as well as the observational data from ground-based
GNSS, space-based GNSS RO, and radiosondes used in this study. We also outline the PWV retrieval method, data quality
control procedures, and the spatio-temporal co-location strategy.

### 2.1 Typhoon data

The typhoon data used in this study are sourced from the typhoon track real-time release system, operated by the Zhejiang
provincial department of water resources and the Zhejiang water resources information management center
(https://typhoon.slt.zj.gov.cn/, last accessed: 27 August 2025). The system provides typhoon center location, time, wind
speed, and additional information, with a temporal resolution of 1–3 hours that becomes finer with increasing typhoon
intensity. The typhoon categories in this dataset follow the classification scheme of the China Meteorological Administration
(CMA), which defines six categories based on wind speed: tropical depression (TD, 10.8–17.1 m/s), tropical storm (TS,
17.2–24.4 m/s), severe tropical storm (STS, 24.5–32.6 m/s), typhoon (TY, 32.7–41.4 m/s), severe typhoon (STY, 41.5–
50.9 m/s), and super typhoon (Super TY, ≥51.0 m/s). For simplicity, these are denoted as L1 to L6, respectively. In this study,
L1 typhoons are not included since no GNSS stations are co-located with their tracks.



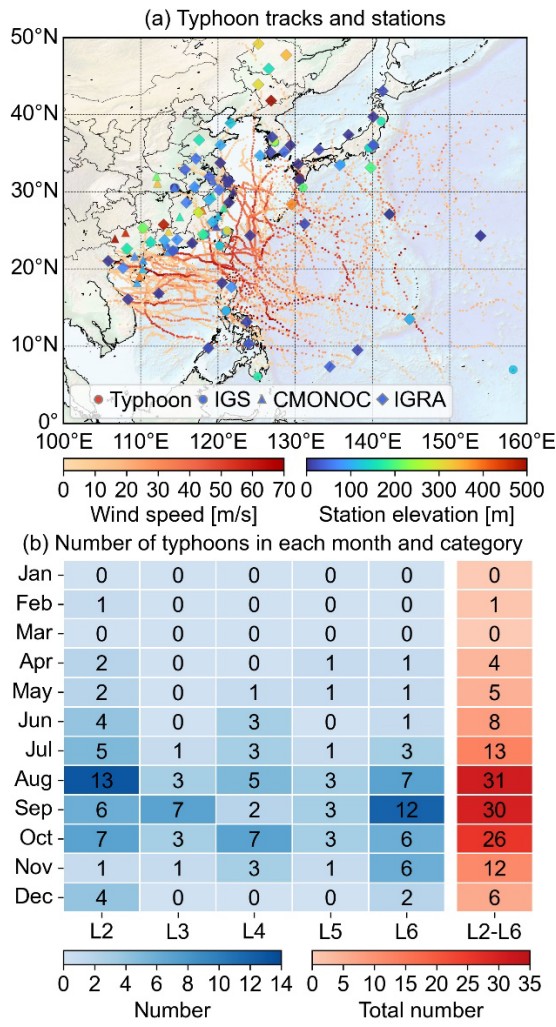

**Figure 1: (a) Spatial distribution of typhoon tracks, ground-based GNSS station from IGS (marked by circles) and CMONOC**
**(marked by triangles), and IGRA radiosonde stations (marked by diamonds) with colorbars indicating wind speed (in m/s, left)**
**and station elevation (in m, right). (b) Number of typhoons in each month and category from January 2020 to December 2024.**

There are 113 typhoons recorded from January 2020 to December 2024 (This period represents the full extent of the available CMONOC dataset), and the spatial distribution of their tracks are shown in Fig.1a. Fig. 1b illustrates the number of typhoons by category over the five-year period, totaled per month. Typhoons occur most frequently from July to November,
with the highest number of typhoons in August, September, and October. Specifically, tropical storms (L2) dominate in August and super-typhoons (L6) in September, while typhoon categories are more evenly distributed in October.



### 2.2 Reanalysis data

#### 2.2.1 ERA5

ERA5 is the fifth generation ECMWF reanalysis for the global climate and weather for the past eight decades from 1940, and is a state-of-the-art reanalysis dataset developed by ECMWF (Hersbach et al., 2020a; Soci et al., 2024). ERA5 is produced using a four-dimensional variational (4D-Var) data assimilation system and model forecasts in CY41R2 of the ECMWF Integrated Forecast System (IFS), with 137 hybrid pressure levels in the vertical and the top level at 0.01 hPa. Atmospheric data are available on the interpolated 37 pressure levels. The native horizontal resolution is 0.25°× 0.25°, and the finest temporal resolution is 1 hour.

#### 2.2.2 MERRA-2

MERRA-2, developed by the Global Modeling and Assimilation Office (GMAO) at the National Aeronautics and Space Administration (NASA), provides global atmospheric reanalysis data starting from 1980 (Gelaro et al., 2017b). Compared to its predecessor MERRA, MERRA-2 incorporates improvements in the Goddard Earth Observing System (GEOS) model and the Gridpoint Statistical Interpolation (GSI) assimilation system, enabling the assimilation of modern satellite observations such as hyperspectral radiance, microwave sensors, RO, and NASA ozone profiles. MERRA-2 offers a horizontal resolution of 0.5° × 0.625°, 72 vertical levels up to 0.01 hPa, and the instantaneous 3-hourly data are used in this study.

#### 2.2.3 JRA-3Q

JRA-3Q is produced by the Japan Meteorological Agency (JMA) using its advanced global numerical weather prediction (NWP) system to enhance the quality and temporal coverage of long-term reanalysis (Kosaka et al., 2024). It builds upon developments since JRA-55 and extends the reanalysis period back to September 1947, covering an earlier era that includes many notable typhoon events. JRA-3Q assimilates a wide range of reprocessed observational datasets, including rescued historical observations and satellite data provided by global meteorological and satellite centers. It employs a 4D-Var data assimilation system and addresses many of the limitations found in JRA-55, resulting in a high-quality and consistent dataset spanning over 75 years. The vertical structure includes 45 pressure levels, with a horizontal resolution of 0.375° × 0.375° and a temporal resolution of 6 hours. It is noteworthy that the JRA-3Q dataset was officially released in 2022, and related evaluation studies are still relatively limited. The evaluation conducted in this study may serve as a helpful reference for its future application.

Table 1 summarizes the data centers, resolutions, start time, update frequency, and assimilation strategies of the three reanalysis datasets. For more detailed information, readers are referred to the publications (Gelaro et al., 2017a; Hersbach et al., 2020b; Bell et al., 2021; Kosaka et al., 2024).



**Table 1. Summary of three atmospheric reanalysis datasets used in this study.**

| Data description | ERA5 data on pressure levels | MERRA-2 M2I3NVASM | JRA-3Q isobaric analysis fields |
|---|---|---|---|
| Organizations | ECMWF | NASA GMAO | JMA |
| Horizontal resolution (lon × lat) | 0.25°×0.25° | 0.625°×0.5° | 0.375°×0.375° |
| Vertical pressure levels | 37 | 72 | 45 |
| Temporal resolution | 1-hourly | 3-hourly | 6-hourly |
| Temporal coverage | Jan 1940–present | Jan 1980–present | Sep 1947–present |
| Update frequency | Daily | Monthly | Monthly |
| Assimilation strategy | 4D-Var | 3D-Var | 4D-Var |

### 2.3 GNSS data

**2.3.1 Ground-based GNSS data**

Ground-based GNSS provides continuous, all-weather, high-precision observations of atmospheric variables with high temporal resolution, making it well-suited for evaluating reanalysis data under typhoon conditions. The International GNSS Service (IGS) provides zenith path delay (ZPD, also known as ZTD) products at 5-minute intervals. However, most IGS stations in the Asia–Pacific region is concentrated in Japan and South Korea, with sparse coverage along China's

southeastern coast. To enhance spatial coverage, we incorporate GNSS data from the Crustal Movement Observation Network of China (CMONOC). These data are processed using the Position and Navigation Data Analyst (PANDA) software developed by Wuhan University (Shi et al., 2008), based on the Precise Point Positioning (PPP) technique (Zumberge et al., 1997), to generate ZTD estimates at the same temporal resolution as the IGS products.

Additionally, a quality control procedure excludes loosely constrained ZTD estimates, defined as those deviating from the

170 station's monthly mean by more than four standard deviations (STD) (Zhang et al., 2017). Fig. 1a shows the locations of IGS and CMONOC stations and their elevation. To minimize errors in the vertical interpolation of PWV, the analysis excludes stations with elevations greater than 500 m. In total, this study uses 34 IGS and 30 CMONOC stations.

### 2.3.2 GNSS RO data

GNSS RO data of the Constellation Observing System for Meteorology, Ionosphere, and Climate-2 (COSMIC-2) mission, as

the successor to COSMIC, are provided via the COSMIC Data Analysis and Archive Center (CDAAC) in Boulder, USA (Schreiner et al., 2020). The orbital inclination of the COSMIC-2 constellation is specifically designed to enhance the number of RO observations over tropical and subtropical regions, resulting in nearly all RO profiles being distributed within a latitude range of 45°N to 45°S. Computing PWV requires atmospheric specific humidity or water vapor partial pressure profiles. This study uses near real time wet profiles (hereafter "wetPrf") from the Level 2 products. The wetPrfs provide

atmospheric parameters with a vertical sampling of 50 m below 20 km and 100 m between 20 km and 60 km (the upper limit



of the profiles). These data are retrieved using a one-dimensional variational (1D-Var) technique, and the lowermost height varies among profiles (Wee et al., 2022). In tropical and subtropical regions, super-refraction often prevents signals from penetrating to the surface, resulting in variation in the lowermost height across COSMIC-2 profiles (Schreiner et al., 2020; Wang et al., 2022). To reduce vertical interpolation errors, this study uses only wetPrfs that pass the CDAAC quality control
procedures and reach below 500 m.

## 2.4 Radiosonde data

Radiosonde data used in this study are obtained from the Integrated Global Radiosonde Archive (IGRA), with routine observations conducted twice daily at approximately 0000 and 1200 UTC. PWV derived from radiosonde profiles typically has an uncertainty of 5% to 8% (Pérez-Ramírez et al., 2014; Turner et al., 2003). Despite certain limitations, radiosonde
observations remain a standard reference for evaluating the PWV retrieved from other techniques (Gui et al., 2017). Quality control is performed following the approaches proposed in previous studies (Wang and Zhang, 2008; Zhang et al., 2017), with additional criteria applied to ensure profile completeness and temporal coverage: (1) Humidity profiles must extend to at least 300 hPa and include measurements at the surface and at a minimum of five standard pressure levels above the surface, regardless of whether surface pressure is above or below 1000 hPa.; (2) Profiles containing large data gaps, defined as
pressure intervals exceeding 200 hPa between successive humidity measurements, are discarded; (3) Stations must operate continuously from January 2020 to December 2024, with at least 200 observations per year; (4) Stations with elevations exceeding 500 m are excluded (Shi et al., 2025). Only profiles with more than 30 vertical levels are used for PWV calculation. Based on these criteria, a total of 60 radiosonde stations is retained for evaluation. The locations of the radiosonde stations are indicated in Fig. 1.

## 2.5 Ground-based station selection scheme and RO co-location method

The gale-force wind radius (R34, where 34 refers to wind speed in knots) is a key parameter for quantifying the spatial extent of a typhoon's impact. Previous studies report that R34 typically ranges from 210 km to 340 km (Sampson et al., 2017; Kim et al., 2022). In this study, the distance between each ground-based station and the typhoon center is calculated, and a station is considered to be in the typhoon area if the minimum distance is less than 300 km. For the comparison using RO
profiles, spatio-temporal co-location with typhoon centers is required. Considering the horizontal smearing of RO profiles, a matching window of 100 km and 30 minutes is applied to associate RO profiles with corresponding typhoon centers.

## 2.6 PWV estimation

PWV can be derived using two approaches. The first integrates specific humidity or water vapor partial pressure profiles and is applied to reanalysis, radiosonde, and RO data. The second converts the zenith wet delay (ZWD) estimated from GNSS
PPP into PWV using a conversion factor.



### 2.6.1 Reanalysis-PWV, radiosonde-PWV, and RO-PWV

The PWV is obtained from vertical integration of the specific humidity ( $q$ , in g·kg⁻¹) profile, expressed as:

$$PWV = \int_{p_1}^{p_2} \frac{q}{\rho_w \cdot g_s} \, dp \tag{1}$$

where $p_1$ and $p_2$ (in hPa) represent the upper and lower pressure boundaries of the integration, respectively, and $g_s$ 

denotes the mean gravitational acceleration, can be written as:

$$g_s(\varphi,h) = g_n \left(1 - 0.0026373\cos(2\varphi) + 5.9 \cdot 10^{-6}\cos^2(2\varphi)\right) \cdot \left(1 - 3.14 \cdot 10^{-7} \cdot h\right) \tag{2}$$

where $g_n = 9.80665$ m·s⁻² is the standard gravitational acceleration, $\varphi$ (in rad) and $h$ (in m) are latitude and elevation, respectively. When water vapor partial pressure $e$ (in hPa) is used instead of specific humidity, the conversion between $e$ and $q$ follows:

$$e = \frac{qp}{0.622 + 0.378q} \tag{3}$$

where $p$ is atmospheric pressure (in hPa).

### 2.6.2 GNSS-PWV

The ZTD, which consists of the zenith hydrostatic delay (ZHD) and ZWD components, can be accurately derived from GNSS observations processed in PPP mode. The ZHD can be precisely calculated using the Saastamoinen model

(Saastamoinen, 1972; Elgered et al., 1991):

$$ZHD = \frac{0.002277 \cdot p_s}{1 - 0.00266 \cdot \cos(2\varphi) - 0.00028 \cdot H} \tag{4}$$

where $\varphi$ is the station latitude (in rad), $H$ denotes the ellipsoidal height (in km), and $p_s$ is the surface pressure (in hPa). The ZWD can be obtained as the difference between ZTD and ZHD:

$$ZWD = ZTD - ZHD \tag{5}$$

the PWV is subsequently computed by scaling ZWD with a water vapor conversion coefficient:

$$PWV = \Pi \times ZWD \tag{6}$$

where $\Pi$ is determined using the following equation:



$$\Pi = \frac{1}{10^{-6} \rho_w R_v \left[ (k_3 / T_m) + k_2' \right]} \tag{7}$$

where $\rho_w = 1000 \ \mathrm{kg \cdot m^{-3}}$ is the density of liquid water, $R_v = 461.51 \ \mathrm{J \cdot K^{-1} \cdot kg^{-1}}$ is the specific gas constant for water vapor, $k_2' = 17 \pm 10 \ \mathrm{K \cdot hPa^{-1}}$ and $k_3 = 3.776 \pm 0.004 \times 10^5 \ \mathrm{K^2 \cdot hPa^{-1}}$ are atmospheric refractivity constants. The parameter $T_m$ represents the weighted mean temperature. It can be obtained either from an empirical linear model between surface temperature and $T_m$ (Bevis et al., 1994) or from vertical integration using meteorological data. Previous comparisons have shown that the integration approach generally yields higher accuracy (Wang et al., 2005). The computation of $T_m$ follows this equation (Davis et al., 1985; Bevis et al., 1992):

$$T_m = \frac{\int_{h_s}^{\infty} (e/T) \, dh}{\int_{h_s}^{\infty} (e/T^2) \, dh} = \frac{\sum_1^n \overline{\left(\frac{e_i}{T_i}\right)}(h_i - h_{i-1})}{\sum_1^n \left(\frac{e_i}{T_i^2}\right)(h_i - h_{i-1})} \tag{8}$$

here, $e$ denotes the water vapor pressure at the station's zenith (in hPa), $T$ is the temperature (in K), and $h$ is the height (in m).

### 2.6.3 PWV vertical adjustment model

To adjust PWV estimated from reanalyses to the height of GNSS or radiosonde stations, and to adjust RO-PWV to the required height for comparison, a vertical adjustment is necessary. The exponential PWV adjustment model is currently widely used and computationally efficient, with the formula as follows:

$$PWV_1 = PWV_2 \cdot \exp\left(-(h_1 - h_2)/2\right) \tag{9}$$

where $PWV_1$ and $PWV_2$ represent the PWV at $h_1$ and $h_2$.

### 2.6.4 Statistical metrics

The statistical metrics used in this study include the systematic deviation of PWV from reanalysis with respect to observations, denoted as bias, and root mean square error (RMSE). To provide a more intuitive representation of the PWV bias relative to the reference value, the relative bias (RB) is defined as follows:

$$RB = \frac{PWV_{reanalysis} - PWV_{reference}}{PWV_{reference}} \times 100\% \tag{10}$$

where $PWV_{reanalysis}$ represents the PWV from different reanalysis data, and $PWV_{reference}$ is the reference PWV, which can be GNSS-PWV, radiosonde-PWV, or RO-PWV.



# 3 Results

This section presents the evaluation results of PWV estimates from the three reanalysis datasets. For clarity and consistency, prefixes are used to distinguish PWV derived from different data sources and their corresponding evaluation metrics. Specifically, GNSS-PWV, RS-PWV, and RO-PWV denote PWV retrieved from ground-based GNSS, radiosonde, and RO data, respectively. Similarly, E-PWV, M-PWV, and J-PWV refer to PWV estimated from ERA5, MERRA-2, and JRA-3Q reanalysis datasets, respectively. The term REA-PWV collectively refers to all reanalysis-derived PWV values. The same naming convention is also applied to evaluation metrics such as bias and RMSE.

## 3.1 Evaluation using ground-based GNSS data

For ground-based GNSS, subscripts are used when necessary to distinguish data from different networks. Specifically, GNSS-PWV$_I$ and GNSS-PWV$_C$ refer to PWV estimated from IGS and CMONOC stations, respectively.

### 3.1.1 Monthly evaluation

To provide a comprehensive understanding of the PWV accuracy of the three reanalysis datasets, evaluations are conducted using data from 64 GNSS stations spanning January 2020 to December 2024. Monthly mean PWV from the reanalyses is compared to GNSS-PWV, with results shown in Fig. 2. The top, middle, and bottom rows of Fig. 2 display the monthly mean PWV (a1–a3), bias and RMSE (b1–b3), and RB (c1–c3), respectively. Red and yellow denote results referenced to GNSS-PWV$_C$, while blue and green correspond to GNSS-PWV$_I$.

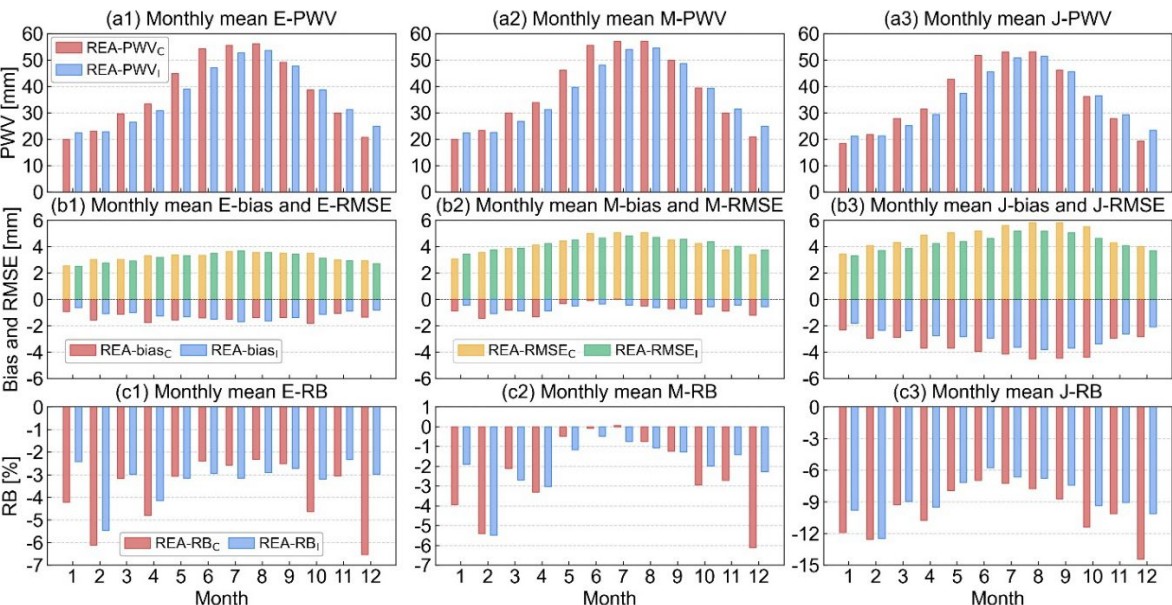

**Figure 2: Monthly mean evaluation results. (a1-a3) represent REA-PWV (in mm); (b1-b3) represent bias and REA-RMSE (in mm); (c1-c3) represent REA-RB. Red and yellow denote results referenced to GNSS-PWV$_C$, while blue and green correspond to GNSS-PWV$_I$.**





The monthly-mean PWV from reanalyses, E-PWV, M-PWV, and J-PWV, exhibits high consistency and pronounced seasonal variation, with peaks of typically about 50 mm in July and August and low values of about 20 mm in boreal winter. The peak of PWV precedes the typhoon season, which occurs from August to October. Overall, GNSS-PWV$_C$ is generally higher than GNSS-PWV$_I$ from February to September, but lower in the other months. The monthly mean E-PWV and M-PWV are similar, while J-PWV remains consistently lower than both across all months. E-PWV and J-PWV show negative biases every month, with the absolute value of the bias positively correlated with PWV, consistent with previous studies indicating that mean REA-PWV is negatively biased in low-latitude regions (Wang et al., 2020). The largest E-bias$_C$ and E-bias$_I$ occur in October and July, with −1.82 mm and −1.63 mm, respectively, while the largest J-bias$_C$ and J-bias$_I$ both occur in August with −4.51 mm and −3.79 mm, respectively, from the GNSS ground-based observation. M-PWV shows good consistency with GNSS-PWV, with M-biases staying below 1 mm to near-zero from May to September, when PWV is relatively high. The monthly mean RMSE follows a similar distribution pattern as PWV, with E-RMSE being the smallest, followed by M-RMSE, and J-RMSE being the largest. RB values are generally smaller in summer and larger in winter. When monthly mean PWV exceeds 40 mm (May to September), despite larger biases, RB remains below 3%.

For the months with more than ten typhoons within the five-year period (July to November), the weighted mean bias and RMSE are calculated based on the number of CMONOC and IGS stations. The E-bias, M-bias, and J-bias are −1.38 mm, −0.58 mm, and −3.74 mm, respectively, while the E-RMSE, M-RMSE, and J-RMSE are 3.38 mm, 4.51 mm, and 5.10 mm, respectively. Among the three datasets, Merra-2 shows the least systematic deviation while ERA5 shows the lowest RMSE, suggesting greater stability for E-PWV. For JRA-3Q, both the absolute value of J-bias and J-RMSE are the largest in most months, The next section presents a more detailed evaluation of REA-PWV during typhoon events.

### 3.1.2 Composite evaluation considering adjacent periods of typhoons

In general, typhoon monitoring agencies release typhoon data based on wind speed. Data recording begins or ends when the wind speed reaches or falls below a specified threshold. However, it has been found that water vapor plays a critical role in TC formation. High column water vapor appears near the pouch center and starts to increase about 42 hours prior to genesis, while a substantial increase in precipitation occurs within 24 hours before genesis (Wang, 2014; Wang and Hankes, 2016). Moreover, even after typhoon dissipation or passage, residual circulation can continue to exert influence (Duan et al., 2014). Therefore, in addition to the recorded typhoon period (hereafter "r-typhoon"), we also evaluate the accuracy of REA-PWVs during the one-week adjacent period (AP), defined as the week before and the week after r-typhoon.

Results are presented in Fig. 3. A schematic of the timeline with r-typhoon and the AP is shown in the top panel of Fig. 3. When referring to AP with a duration of X days, it indicates a period covering X days before and after r-typhoon. Figure 3 (a1–c4) presents the results for ERA5, MERRA-2, and JRA-3Q, respectively, while Fig. 3 (a3) and (a4) indicate the number of station-typhoon pairs. On the x-axis, L2 to L6 represent different typhoon categories, "L2–6" denotes all typhoon categories, and "Non" refers to the mean results for the same stations and periods in non-typhoon years during 2020–2024.





For example, in the case of typhoon Doksuri (July 21–29, 2023), "Non" refers to results from the same station and the same period in 2020, 2021, 2022, and 2024, when no typhoon occurred.

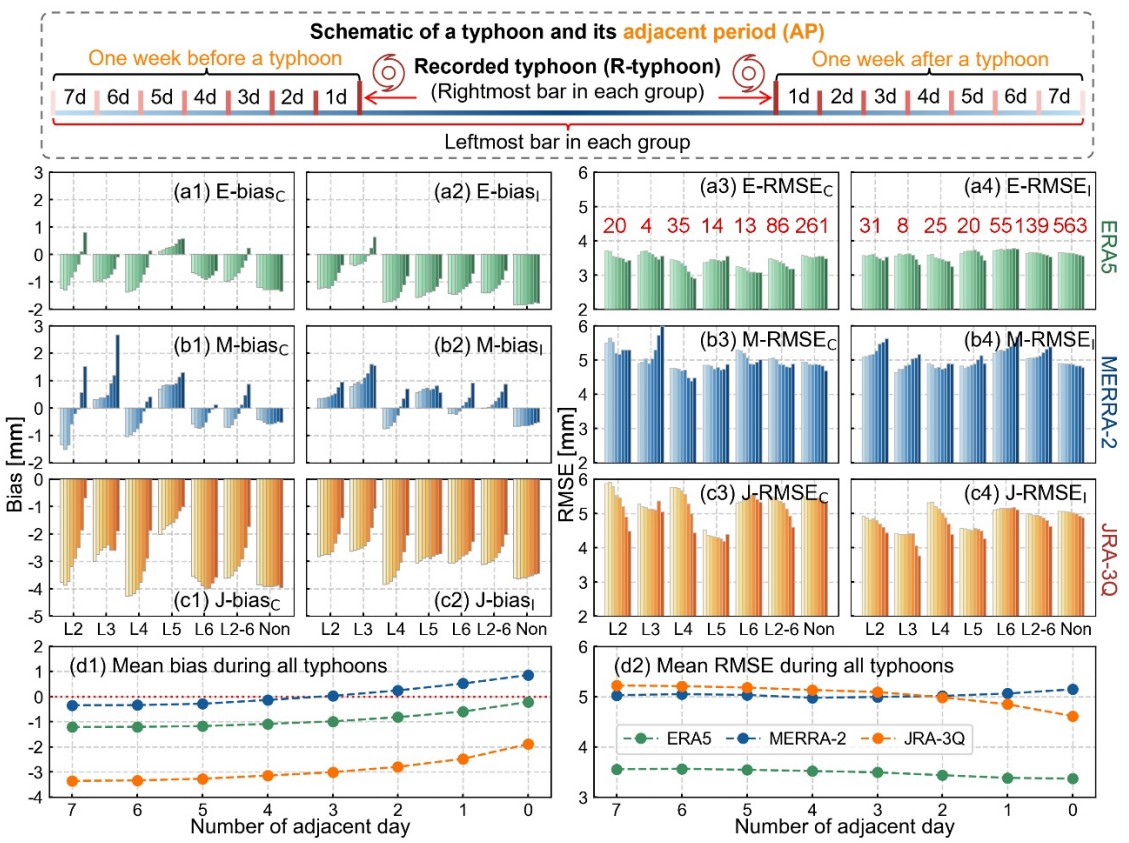

**Figure 3:** The top panel shows a schematic of a typhoon, including the r-typhoon and AP. The bottom panel presents REA-biases and REA-RMSEs (in mm) for different typhoon categories, with AP ranging from 7 days before/after to 0 day (color from light to dark). The x-axis shows typhoon categories or conditions. Panels (a3) and (a4) show the number of station-typhoon pairs, while (d1) and (d2) display the mean REA-biases and REA-RMSEs for all typhoons.

For biases (Fig. 3, left panels), all three REA-biases generally decrease as the AP shortens. Specifically, the absolute values of E-bias and J-bias decrease, while that of M-bias increases. E-bias$_C$ and E-bias$_I$ (except during L6 typhoons) tend to decrease as AP shortens. For L2–L4 typhoons, both E-bias$_C$ and E-bias$_I$ change from negative to positive as AP shortens, with this effect most pronounced for E-bias$_C$ during L5 typhoons. According to Fig. 3 (d1), all E-biases are smaller than 1 mm, indicating good agreement between E-PWV and GNSS-PWV even when the bias turns positive. M-bias is predominantly positive and becomes more pronounced as AP shortens, though the mean bias for all typhoons remains within 1 mm. For L3 typhoons, the M-bias$_C$ exceeds 2.5 mm, which may be due to the limited sample size of only four events. All J-biases are negative. For L3 and L6 typhoons, J-bias$_C$ does not decrease monotonically as AP shortens. Overall, the absolute value of J-bias is the largest. Figure 3 (d1) shows that both E-bias and J-bias decrease as AP shortens, with values within r-





typhoon of –0.21 mm and –1.89 mm, respectively. The absolute value of M-bias decreases from 7 adjacent days to 3
adjacent days, but becomes positive and increases when the AP is less than 3 days.

For RMSE (Fig. 3, right panels), both E-RMSE and J-RMSE tend to decrease with a shorter AP, especially for E-RMSE in
L4 typhoons and J-RMSE in L2 and L4 typhoons. M-RMSE$_C$ shows no clear pattern with AP changes, whereas M-RMSE$_I$
generally increases as AP shortens. According to Fig. 3 (d2), E-RMSE, M-RMSE, and J-RMSE for all typhoon periods
remain nearly unchanged with AP variations. M-RMSE and J-RMSE are similar near 5 mm, while E-RMSE is the lowest
near 3.5 mm.

Under non-typhoon conditions, REA-biases and REA-RMSEs remain nearly unchanged with varying AP. Without
considering AP, E-bias, M-bias, and J-bias are –1.58 mm, –0.53 mm, and –3.69 mm, respectively; E-RMSE, M-RMSE, and
J-RMSE are 3.51 mm, 4.73 mm, and 5.08 mm, respectively. Among the three datasets, ERA5 yields the lowest RMSE,
MERRA-2 shows the smallest bias, while JRA-3Q exhibits the largest bias and large RMSE.

### 3.1.3 Evaluation during recorded typhoon periods

This section focuses on the accuracy of PWVs during r-typhoon from reanalyses, without considering AP. Table 2 presents
mean REA-biases, REA-RMSEs and dRMSEs of all typhoons during both r-typhoon and non-typhoon. In addition, to
distinguish between systematic bias and random error, the de-biased RMSE (dRMSE) is introduced as a supplementary
metric to reflect the random component after removing bias. It should be noted that dRMSE is used as a reference metric to
understand the errors better, while bias and RMSE remain the primary metrics for subsequent evaluation.

Table 2. Mean REA-biases, REA-RMSEs, and REA-dRMSEs (in mm) of all typhoons during r-typhoon and non-typhoon periods.

| Metrics | Reanalyses | R-typhoon | Non-typhoon |
|---------|-----------|-----------|-------------|
| Bias | ERA5 | –0.29 | –1.65 |
| | MERRA-2 | 0.86 | –0.53 |
| | JRA-3Q | –1.92 | –3.61 |
| RMSE | ERA5 | 3.40 | 3.52 |
| | MERRA-2 | 5.19 | 4.74 |
| | JRA-3Q | 4.61 | 5.01 |
| dRMSE | ERA5 | 2.86 | 2.58 |
| | MERRA-2 | 4.55 | 4.01 |
| | JRA-3Q | 3.57 | 2.84 |

For biases, E-biases and J-biases during r-typhoon periods are smaller in comparison to the non-typhoon periods across all
typhoon categories, while the M-bias is only smaller for the L4 category during r-typhoon. For all typhoons, the mean biases
during non-typhoon periods are –1.65 mm and –3.61 mm for E-bias and J-bias, respectively, and –0.29 mm and –1.92 mm
during r-typhoon periods, respectively. M-bias changes from –0.53 mm to 0.86 mm, increasing in absolute value and shifting
from negative to positive. These results indicate that the overall underestimation of E-PWV and J-PWV is alleviated during





r-typhoon, while M-PWV shifts from underestimation during non-typhoon to overestimation during r-typhoon, with the overestimation being more evident for L2 and L3 typhoons.

For RMSEs, both E-RMSE and J-RMSE are generally lower during r-typhoon compared to non-typhoon, decreasing by 0.12
350  mm and 0.40 mm, respectively. In contrast, M-RMSE increases by 0.45 mm during r-typhoon. The REA-dRMSEs for all categories and individual typhoon levels are higher during r-typhoon compared to non-typhoon. These results indicate that E-RMSE is the lowest and least affected by typhoons, whereas M-PWV exhibits higher uncertainty during typhoons. Interestingly, the REA-dRMSEs for all categories and individual typhoon levels are higher during r-typhoon compared to non-typhoon, opposite to REA-RMSE, which is lower during r-typhoon. This shows that the larger REA-RMSEs during
non-typhoon periods, especially for E-RMSEs and J-RMSEs, are not caused by an increase in random error – which is indeed decreasing – but by an increase in bias. Thus, during typhoons, for ERA5 and JRA-3Q, the random error is higher, but the bias is much lower, leading to lower overall error.

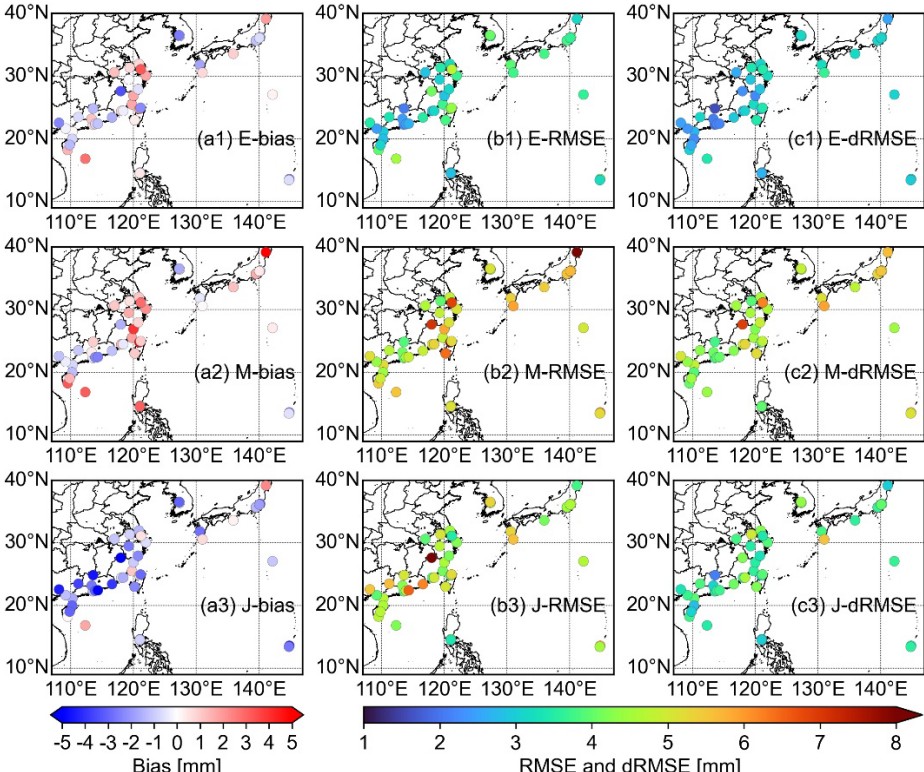

**Figure 4: Spatial distribution of mean REA-biases, REA-RMSEs, and REA-dRMSEs (in mm) at GNSS station locations during r-
typhoon for all typhoons.**

Fig. 4 shows the spatial distribution of REA-biases, REA-RMSEs, and REA-dRMSEs at GNSS stations during r-typhoon period for all typhoons. For biases, E-bias and M-bias exhibit similar spatial patterns: most stations in southern China and South Korea have negative biases, while those in eastern China are mainly positive. Notably, some stations in South China Sea, Philippines, Southern Japan also show prominent positive biases. J-bias is predominantly negative, with only a few



stations showing positive values. For RMSE, E-RMSE is generally the lowest and most uniformly distributed, while M-RMSE is higher and shows relatively large values (>8 mm) at a station located in Southern Japan. J-RMSE falls between the other two, with the highest value (8.12 mm) observed at a station located in Fujian province, China. For dRMSE, E-dRMSE is the lowest overall, followed by J-dRMSE, while M-dRMSE is the highest and exhibits a clear latitudinal dependence, being larger at higher latitudes. Additionally, JRA-3Q displays the largest difference between RMSE and dRMSE among the

three datasets, further indicating that the relatively large absolute J-biases substantially impact J-RMSE.

To sum up, E-PWVs and J-PWVs show improved accuracy during r-typhoon compared to non-typhoon, while M-PWVs exhibit a slight decrease in accuracy. However, the absolute value of mean M-bias remains within 1 mm, and its mean RMSE increases by only 0.45 mm. Despite some deterioration, M-PWVs still maintain a comparable level of accuracy during typhoons. These results indicate that E-biases and J-biases are lower and their stability is better under the dynamic

conditions of typhoons.

Nevertheless, the underlying causes of the changes in accuracy for all three REA-PWVs during typhoons warrant further investigation. JRA-3Q assimilates JMA tropical cyclone bogus (TCB) data to improve the accuracy of typhoon analysis (Kosaka et al., 2024). This assimilation provides prior information on typhoons and helps constrain the estimation of atmospheric parameters, resulting in reduced J-bias and J-RMSE during typhoons. MERRA-2 does not assimilate any

estimates of TC central surface pressures. Instead, TCs detected in the model background fields are relocated using the position given in the NCEP tcvitals reports following an established method (Liu et al., 2000; Koster et al., 2016), which enables MERRA-2 to more accurately capture typhoon processes. Despite this, M-PWV is generally overestimated and M-RMSE increases during typhoons.

Regarding RMSE, the reductions in E-RMSEs and J-RMSEs during r-typhoon are smaller than the corresponding

improvements in E-biases and J-biases. This may be attributed to substantial atmospheric variability during typhoons, which maintains high random PWV uncertainties even with data assimilation. Overall, this study focuses on evaluating the accuracy of the three REA-PWVs during typhoons, while the underlying causes of changes in their accuracy merit further investigation.

### 3.2 Evaluation using radiosonde observations

Although all three reanalysis datasets assimilate radiosonde observations, those observations are used as a reference due to their high accuracy, serving to further evaluate REA-PWVs. Fig. 5 presents the evaluation results in terms of correlations between REA-PWVs and RS-PWVs. According to Fig. 5 (a–c), all three REA-PWVs exhibit high agreement with RS-PWVs during pre-typhoon, r-typhoon, and post-typhoon, especially for ERA5, whose correlation coefficients remain above 0.9. The correlation is slightly lower during pre-typhoon compared to r-typhoon and post-typhoon. Based on Fig. 5 (d–h), the

correlation between REA-PWVs and RS-PWVs does not show notable variation across different categories, except for a noticeable decrease during L5 typhoons, where the correlation coefficients for ERA5 and MERRA-2 are below 0.9 and 0.8, respectively, and that for JRA-3Q also reaches its lowest value, 0.86.





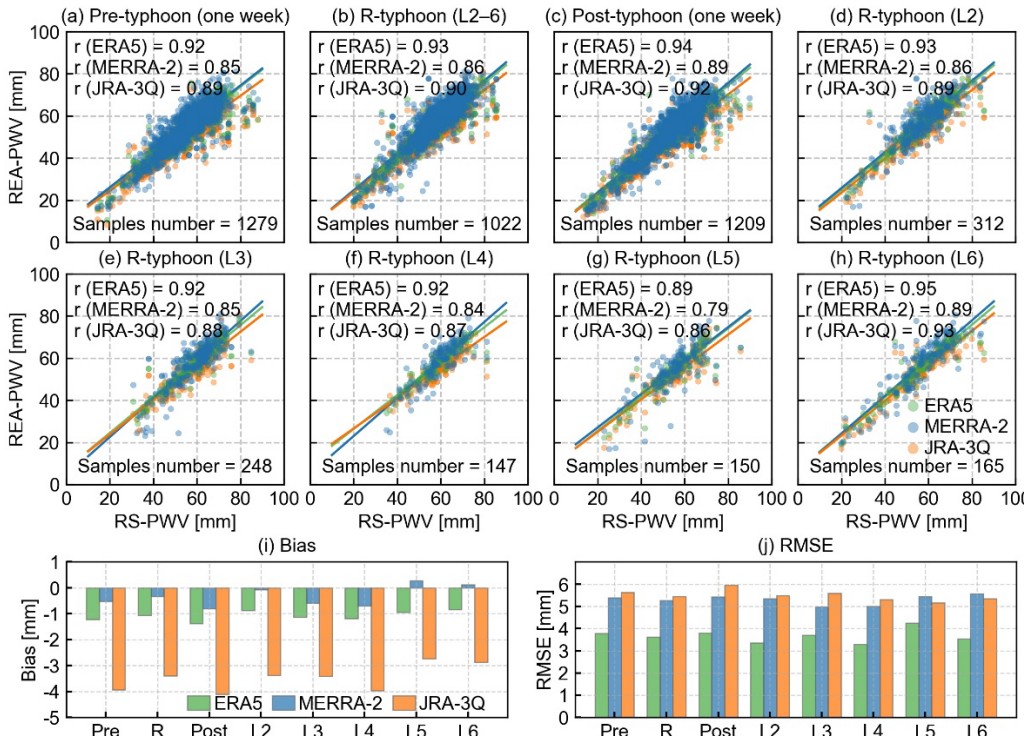

**Figure 5: Correlation statistics of REA-PWVs and RS-PWVs during pre-typhoon (one week before r-typhoon) (a), r-typhoon (b), and post-typhoon (one week after r-typhoon) (c), as well as during r-typhoon for different categories (d–h). Panels (i) and (j) display bar charts of REA-biases and REA-RMSEs corresponding to Fig. 5 (a–h). In the figure, "r" denotes the correlation coefficient, and the number of samples is also showed.**

The results show that nearly all biases are negative, with positive M-biases observed only for L5 and L6 typhoons, both less than 0.3 mm. The absolute value of M-bias is the smallest, followed by E-bias, while J-bias has the largest absolute value. The mean REA-biases during r-typhoon are lower than those during pre-typhoon and post-typhoon. For RMSE, E-RMSE is consistently the lowest, M-RMSE exceeds J-RMSE during L5 and L6 typhoons but is lower than J-RMSE for other categories. The variations of REA-RMSEs among pre-typhoon, r-typhoon, and post-typhoon are generally small. Notably, only J-RMSE increases by 0.3–0.5 mm in post-typhoon compared to pre-typhoon and r-typhoon. All three REA-RMSEs reach their lowest values during r-typhoon. Moreover, although the overall accuracy of J-PWV is lower than that of M-PWV, the correlation coefficients for J-PWV are higher than those for M-PWV in all cases (a–h), which may be due to systematic biases. These results indicate that all three REA-PWVs maintain strong correlations with RS-PWV under typhoon conditions.

### 3.3 Comparison with COSMIC-2 RO profiles

Evaluation using GNSS-PWVs and RS-PWVs as references is limited by station locations, whereas RO-PWVs can be used to compare REA-PWVs over oceanic regions. Based on the co-location scheme described in Section 2.5, there are 49 and



216 COSMIC-2 RO profiles co-located with typhoon centers during r-typhoon and non-typhoon, respectively. Fig. 6 presents the comparison results for r-typhoon and non-typhoon. REA-PWVs exhibit strong correlations with RO-PWV, with correlation coefficients of 0.92, 0.86, and 0.87 during r-typhoon. Overall, the correlation coefficients during r-typhoon are slightly lower than those during the non-typhoon.

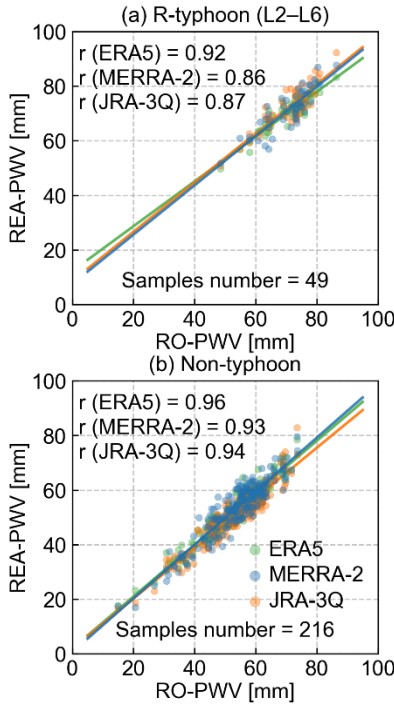

**Figure 6: Correlation statistics of REA-PWVs and RO-PWVs during r-typhoon (a) and non-typhoon (b). In the figure, "r" denotes the correlation coefficient, and the number of samples is also showed.**

**Table 3. REA-biases and REA-RMSEs relative to RO-PWVs during r-typhoon and non-typhoon (in mm).**

|  | R-typhoon | | Non-typhoon | |
|---|---|---|---|---|
|  | Bias | RMSE | Bias | RMSE |
| ERA5 | −0.45 | 2.85 | −0.67 | 2.73 |
| MERRA-2 | 0.67 | 4.05 | −0.34 | 3.84 |
| JRA-3Q | 1.63 | 4.14 | −2.45 | 4.03 |

Table 3 presents the biases and RMSEs of the three REA-PWVs during the r-typhoon and non-typhoon periods. The
425 absolute values of E-bias and J-bias are smaller by 0.22 mm and 0.82 mm during r-typhoon compared to the non-typhoon periods, respectively, while the absolute value of M-bias increases by 0.33 mm. For RMSE, all three REA-PWVs have higher values during r-typhoon than in the non-typhoon period, with increases of 0.12 mm, 0.21 mm, and 0.11 mm, respectively, indicating increased uncertainty in REA-PWVs under typhoon conditions. Although there are more than 5,000 COSMIC-2 RO profiles per day in global tropical and subtropical regions, the number of co-located RO profiles remains



limited under the strict co-location criteria. Therefore, the results in this section are intended primarily as complementary information.

### 3.4 Neighborhood standard deviation of REA-PWVs

Neighborhood standard deviation (NSD) quantifies the local spatial variability of reanalysis data, reflecting the consistency among neighboring grid points (Wei et al., 2013). Higher NSD indicates stronger spatial heterogeneity and larger
representativeness errors when comparing with point observations, which should be considered in evaluation (Bock and Parracho, 2019). NSD is particularly useful for characterizing spatial heterogeneity during extreme weather events such as typhoons, providing insights into the reliability of reanalysis products.

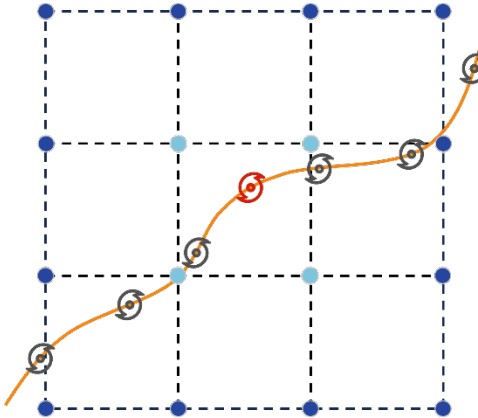

**Figure 7: Schematic of NSD calculation. The orange line shows the typhoon track; the red marker indicates the typhoon center.**
**The 16 nearest grid points are shown in dark blue and light blue, with the 4 closest points in light blue.**

To evaluate local spatial heterogeneity of the three REA-PWVs near typhoon centers, NSD is calculated at two scales based on the standard deviation of PWVs at the 16 nearest grid points (NSD-16) and at the 4 nearest grid points (NSD-4) around the typhoon center, as illustrated in Fig. 7. When calculating NSD, PWV at each grid point is interpolated to the altitude of the typhoon center.

For all typhoons under non-typhoon conditions, E-NSD-16, M-NSD-16, and J-NSD-16 are 2.23, 2.85, and 1.59 mm, while E-NSD-4, M-NSD-4, and J-NSD-4 are 1.24, 1.61, and 0.84 mm, respectively. In both NSD-16 and NSD-4, J-NSD is the smallest, E-NSD is intermediate, and M-NSD the largest, indicating strongest spatial consistency for J-PWV and weakest for M-PWV. During r-typhoon periods, all NSDs increase: E-NSD-16, M-NSD-16, and J-NSD-16 rise to 2.61, 4.17, and 2.30 mm, representing relative increases of 17.1%, 46.4%, and 44.8%, while E-NSD-4, M-NSD-4, and J-NSD-4 rise to 1.62, 2.64,
and 1.31 mm, corresponding to increases of 30.9%, 63.5%, and 56.2%. M-NSD shows the largest absolute and relative increases among the three reanalyses. Overall, the NSDs of the three REA-PWVs increase markedly during typhoons, indicating enhanced spatial heterogeneity under typhoon conditions. Despite this increase, E-NSD and J-NSD generally remain below 3 mm, while M-NSD stays within 5 mm.





A direct evaluation of the NSD is not possible, as NSD cannot be computed from the observations used in this study.
However, the NSD results help in interpreting the results of the errors and biases of the reanalysis products. MERRA has the highest random error (dRMSE) of the three reanalysis datasets. An overestimation of NSD would lead to higher random error. Therefore, this suggests that the NSD of MERRA is overestimated, and the lower NSD of ERA and JRA is closer to the true values – which is in line with the lower dRMSE of ERA and JRA. This also explains why MERRA, despite having generally lower absolute bias levels compared to JRA, still has higher overall RMSE.

## 4 Discussion

Beyond the direct accuracy evaluation, a further aspect to consider is how typhoon center–station distance (hereafter "distance") and wind speed may influence the performance of REA-PWVs, since these factors are not explicitly accounted for when comparing them with GNSS-PWVs. Therefore, this section further investigates the influence of these two parameters. Fig. 8 presents three representative typhoon–station pairs with minimum distances less than 200 km: In-fa–JSLS,
Doksuri–KMNM, and Khanun–AIRA. To facilitate the discussion of water vapor–related changes during each typhoon, precipitation obtained from the Global Precipitation Measurement (GPM) mission is also plotted (gray bars). The right panels show the corresponding time series of REA-biases.

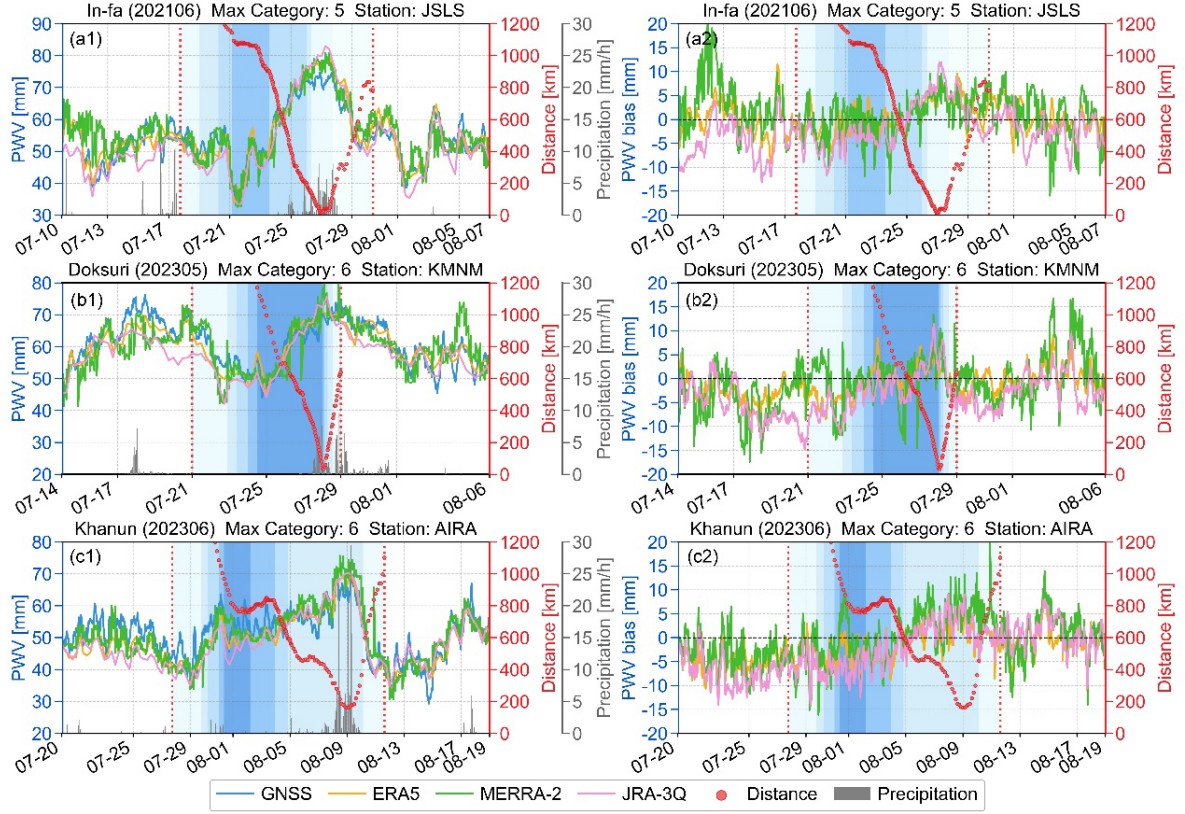





**Figure 8: Time series of PWVs and biases for three typhoon–GNSS station pairs. The left and right columns show PWVs and biases, respectively. PWVs and biases correspond to the left y-axis (in mm); distance to the right red y-axis (in km); and precipitation to the right gray y-axis (in mm/h). Vertical red dashed lines divide the series into pre-typhoon, r-typhoon, and post-typhoon periods. Each panel is annotated with the typhoon number, name, maximum category, and station. Blue shading of varying intensity (darker for stronger categories) indicates typhoon categories, and the x-axis shows month–day.**

According to Fig. 8 (a1–c1), all three REA-PWVs demonstrate generally high consistency with GNSS-PWVs. However, as shown in Fig. 8 (a2–c2), large absolute values of M-biases, sometimes exceeding 10 mm and even approaching 20 mm, are observed during pre-typhoon of In-fa, post-typhoon of Doksuri, and r-typhoon of Khanun. Additionally, J-biases during pre-typhoon also reach negative values less than –10 mm. In contrast, E-biases perform better overall, with fewer occurrences of absolute biases greater than 10 mm. Analysis of the distance pattern reveals that when the distance gets below approximately 600 km, all three REA-biases tend to shift from negative to positive, suggesting a certain correlation between REA-biases and distance. On the other hand, no clear trend is observed in REA-biases with changes in typhoon categories.

Further statistical analysis shows that during r-typhoon for all typhoons, the correlation coefficients between distance and E-bias, M-bias, and J-bias are –0.286, –0.202, and –0.370, respectively, indicating weak negative correlations. Meanwhile, the correlation coefficients with wind speed are all below 0.1, indicating no relationship. These results suggest that, when distance and wind speed are not explicitly considered, the evaluation of REA-PWVs based on GNSS-PWVs is generally reasonable and representative. Although neither distance nor wind speed show a clear correlation with PWV accuracy, the influence of distance appears to be slightly stronger than that of wind speed. This analysis provides a preliminary understanding of the potential influence of distance and wind speed on the evaluation of REA-PWVs. Future work may consider introducing multivariate approaches to enable more detailed evaluations.

## 5 Conclusions

This study provides the first systematic evaluation of the accuracy of PWV estimates from ERA5, MERRA-2, and JRA-3Q reanalysis datasets under typhoon conditions, using ground-based GNSS, radiosonde, and RO observations. More than 100 typhoon events from 2020 to 2024 are examined across four scenarios, namely pre-typhoon, r-typhoon, post-typhoon, and adjacent periods, and results for each scenario are compared with the result during non-typhoon periods.

Evaluation using GNSS-PWVs shows that ERA5 exhibits the most stable performance, with smallest biases and RMSE during typhoons, and slightly larger biases and errors in non-typhoon periods. While being less accurate than ERA5, JRA-3Q also has less bias and RMSE during typhoons than in non-typhoon periods, indicating that the assimilation of TCB observations has a positive contribution to PWV estimation. MERRA-2 has least accuracy in terms of RMSE. In contrast to ERA5 and JRA-3Q, its accuracy is less during typhoons compared to non-typhoon periods. Error decomposition showed that while the RMSE of MERRA-2 is the highest of the three reanalysis datasets, its bias is the smallest during non-typhoon periods, and the second smallest during typhoons. The bulk of its RMSE stems from random error components. This is likely caused by the fact that MERRA-2 shows much higher spatial variability of PWV in the neighborhoods around typhoon



tracks than ERA5 and JRA-3Q, estimated via neighborhood standard deviation. ERA5 and JRA-3Q have similar spatial variability and are thus consistent with each other.

Evaluations using RS-PWVs and RO-PWVs confirm the overall consistency of the three reanalyses with GNSS-based results, supporting their reliability under typhoon conditions. However, since RS and RO samples are limited, these results should be regarded as supplementary, with the GNSS-based evaluation providing the main reference.

Through a comprehensive evaluation, this study demonstrates that reanalysis data can provide continuous and reasonably reliable PWV information under typhoon conditions, even in regions where ground-based observations are sparse or unavailable. These results offer valuable references for water vapor research and practical support for typhoon monitoring and forecasting during extreme weather events. Future work will extend the evaluation to a global scale to assess the performance of various reanalysis water vapor products during TCs worldwide.

**Data Availability**

Typhoon information data are provided by the typhoon track real-time release system, operated by the Zhejiang provincial department of water resources and the Zhejiang water resources information management center (https://typhoon.slt.zj.gov.cn/). GNSS tropospheric products and observations are provided by IGS (https://cddis.nasa.gov/archive/gnss/products/troposphere/zpd/) and CMONOC (data are available upon request and can be downloaded from the FTP server at ftp.cgps.ac.cn after approval). COSMIC-2 RO wet profiles are provided by UCAR (https://data.cosmic.ucar.edu/gnss-ro/cosmic2). ERA5 hourly data on pressure levels is provided by ECMWF (https://cds.climate.copernicus.eu/datasets/reanalysis-era5-pressure-levels). MERRA-2 data is provided by GMAO (https://disc.gsfc.nasa.gov/datasets?project=MERRA-2). JRA-3Q data is provided by JMA (https://rda.ucar.edu/datasets/d640000/). Radiosonde data is provided by IGRA (https://www.ncei.noaa.gov/data/igra). GPM IMERG Final Precipitation L3 Half Hourly 0.1 degree x 0.1 degree V07 data is provided by JAXA and NASA (https://disc.gsfc.nasa.gov/datasets/GPM_3IMERGHH_07/summary?keywords=3IMERGHH).

**Author contributions**

Conceptualization by JS. Data curation was performed by JS, JH, and YF. Methodology was developed by JS, MZ, and YF, with validation provided by JS and WG. Visualization was carried out by JS and JH. Formal analysis was contributed by AKS, MZ, and WG. Software support was provided by ML. Investigation was conducted by JS. Supervision was provided by ML and AKS. Writing – original draft was prepared by JS, and writing – review and editing was contributed by ML, AKS, SS, and KZ. Funding acquisition was secured by ML. All authors discussed the results, contributed to the manuscript revision, and agreed to the final version of the paper.



**Competing interests**

The authors declare that they have no conflict of interest.

**Special issue statement**

This manuscript is submitted to the special issue "The SPARC Reanalysis Intercomparison Project (S-RIP) Phase 2 (ACP/WCD inter-journal SI)".

**Acknowledgments**

This research was partially supported by the Special Scholarship for Graduate Students' Overseas (Outbound) Exchange Program of Wuhan University. Part of the work was carried out during my research stay at the Wegener Center for Climate and Global Change, University of Graz, Austria. The authors would also like to thank Dr. Fan Si from Physikalisches Institut der Ruprecht-Karls-Universität Heidelberg, Germany, Junyi Gao from the University of Edinburgh, UK, and Junyi Han, for their help.

**Financial support**

This study is supported by the Natural Science Foundation of Hubei Province, China (Grant No. 2025AFA038) and the National Natural Science Foundation of China (Grant Nos. 42474032, 42030109, 41931075, and 42004020).

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
