# Peer review of "Evaluation of reanalysis precipitable water vapor under typhoon conditions using multi-source observations"

_EGUsphere, 2025_

## Referee Comment (RC1)

**General comments:**

This study presents a valuable and timely evaluation of PWV (Precipitable Water Vapor or Integrated Water Vapor) from three state-of-the-art reanalysis products (ERA5, JRA-3Q, and MERRA2) within extreme environment of typhoons over Northwest Pacific basin. The authors are to be commended for using a comprehensive suite of multi-source observations (GNSS, radiosonde, RO). In terms of statistical methods, this study uses bias, RMSE and debiased RMSE to compare systematic errors, random errors across three reanalysis under defined four different typhoon scenarios. Overall, these statistics give reasonable conclusions that ERA5 offers the most reliable PWV estimates under typhoon conditions.

However, despite these strengths, the manuscript in its current form has major deficiencies that prevent preclude a recommendation for publication in ACP. These deficiencies include a lack of in-depth analysis to explain the performance differences, several unsubstantiated attributions for the results, inappropriate references and loose connection between the figures and their description in main text. These points are outlined in the detailed comments are shown below.

**Specific comments:**

- C1: In line 34, the statement, "The spatio-temporal variation and distribution of PWV does not only influence the vertical humidity structure", is scientifically imprecise. After examination, the provided citations (Kim et al., 2022; Liu et al., 2023) do not appear to support this specific claim. The authors should revise this sentence for scientific accuracy and ensure that the cited literature directly substantiates the point being made.
- C2: In line 35, the citation to "Kim et al., 2022" is ambiguous. Based on the reference list, this should likely be distinguished as "Kim et al., 2022a" or "Kim et al., 2022b". Please verify and correct this instance and all subsequent citations to this literature.
- C3: In line 36, cyclones should be capitalized to give abbreviations: TCs.
- C4: In line 37, like the C2 for Wang et al., 2020.
- C5: In line 40, the term "translational speed" should be corrected. The standard and more formal term used in the field for the movement of a typhoon is "translation speed."
- C6: In line 40, although the climate trends are observed and modelled, the specific numbers describing the reduction of translation speed and increase of precipitation intensity are not verified in referred literatures. Please check it.

- C7: In line 48, the description of the locations impacted by recent typhoons is geographically inaccurate. The precipitation extremes caused by Typhoon In-fa (2106) and Typhoon Doksuri (2306) located in Henan Province and Beijing-Tianjin-Hebei region correspondingly rather than northern and northeastern China.
- C8: In line 70, the statement explaining the sources of differences among reanalysis datasets is incomplete. It correctly identifies "data assimilation strategies" but omits an equally critical factor: the underlying numerical models themselves.
- C9: In line 80, an ambiguous citation format is used again.
- C10: In line 103, redundant line.
- C11: For a scientific study of typhoon track, it is standard practice to use the official "best track" data (e.g., <a href="https://tcdata.typhoon.org.cn/zjljsjj.html">https://tcdata.typhoon.org.cn/zjljsjj.html</a>).
- C12: In line 130, another ambiguous citation and repeated literatures in refence list.
- C13: In line 131, the description of the ERA5 data assimilation system as simply "four-dimensional variational (4D-Var)" is an oversimplification. For technical accuracy, it should be specified that ERA5 employs a more advanced ensemble 4D-Var system.
- C14: In line 137, similar to C12.
- C15: In line 193, what is the meaning of "a minimum of five standard pressure levels above the surface"?
- C16: In line 214, The term  $\rho_w$  lacks a definition in the main text.
- C17: In Figure 2 legend, the notations REA-PWVc and REA-PWVi are used in the figure and caption but are not defined. While the text defines GNSS-PWV with subscripts for the CMONOC and IGS networks, the use of the reanalysis is confusing.
- C18: In section 3.1.1. Throughout the results section, the discussion of multi-panel figures would be significantly improved by consistently referencing specific subplots (e.g., "Fig. 2a," "Fig. 3d"). Currently, the text makes detailed quantitative statements without nagivating the reader to the evidence, forcing them to search. For example, in Section 3.1.1, the descriptions of bias and RMSE should be explicitly linked to panels (b1-b3) and (c1-c3). Furthermore, a specific instance of this lack occurs on line 343,

where the statement regarding M-bias for L4 typhoons is made without reference to the supporting figure panel. As a general note on presentation, the subplot labeling scheme itself (e.g., 'a1', 'b1') is unconventional; a standard sequential alphabetic scheme (a, b, c, d) is strongly recommended for clarity and adherence to publication norms.

- C19: In line 377 (major scientific concern). The manuscript's central attribution for JRA-3Q's improved PWV accuracy—the assimilation of tropical cyclone bogus (TCB) data—is physically unsubstantiated. As the authors' own reference (Kosaka et al., 2024) states, the TCB data used in JRA-3Q constrains only dynamical fields (sea level pressure and winds) and contains no humidity information. There is no direct pathway for this data to improve the moisture field. The authors must either provide a rigorous, physically-based hypothesis for how the dynamical constraints indirectly improve PWV and support it with further analysis, or this unsubstantiated claim should be removed.
- C20: In line 381, The citations to (Liu et al., 2000; Koster et al., 2016) appear in the text, but the full entries are missing from the reference list. Please add the complete reference details for these sources.
- C21: In line 444, what's the meaning of "altitude of the typhoon center"? In my opinion, the typhoon center is located using latitude and longitude, regardless of height.